# Sphingosine Kinases at the Intersection of Pro-Inflammatory LPS and Anti-Inflammatory Endocannabinoid Signaling in BV2 Mouse Microglia Cells

**DOI:** 10.3390/ijms24108508

**Published:** 2023-05-09

**Authors:** Sara Standoli, Cinzia Rapino, Camilla Di Meo, Agnes Rudowski, Nicole Kämpfer-Kolb, Luisa Michelle Volk, Dominique Thomas, Sandra Trautmann, Yannick Schreiber, Dagmar Meyer zu Heringdorf, Mauro Maccarrone

**Affiliations:** 1Department of Bioscience and Technology for Food Agriculture and Environment, University of Teramo, 64100 Teramo, Italy; sstandoli@unite.it (S.S.); 2Department of Veterinary Medicine, University of Teramo, 64100 Teramo, Italy; crapino@unite.it; 3Institute of General Pharmacology and Toxicology, University Hospital Frankfurt, Goethe University Frankfurt, 60590 Frankfurt am Main, Germany; rudowski@med.uni-frankfurt.de (A.R.); n.kolb@em.uni-frankfurt.de (N.K.-K.); volk@med.uni-frankfurt.de (L.M.V.); 4Institute of Clinical Pharmacology, Goethe University Frankfurt, 60590 Frankfurt am Main, Germany; dominique.thomas@itmp.fraunhofer.de (D.T.); trautmann@med.uni-frankfurt.de (S.T.); 5Fraunhofer Institute for Translational Medicine and Pharmacology (ITMP), Fraunhofer Cluster of Excellence for Immune Mediated Diseases (CIMD), 60596 Frankfurt am Main, Germany; yannick.schreiber@itmp.fraunhofer.de; 6Department of Biotechnological and Applied Clinical Sciences, University of L’Aquila, 67100 L’Aquila, Italy; 7European Center for Brain Research (CERC), Santa Lucia Foundation, Scientific Institute for Research, Hospitalization and Healthcare, 00143 Rome, Italy

**Keywords:** neuroinflammation, microglia, endocannabinoid, fatty acid amide hydrolase, sphingosine kinases, cannabinoid 2 receptor

## Abstract

Microglia, the resident immune cells of the central nervous system, play important roles in brain homeostasis as well as in neuroinflammation, neurodegeneration, neurovascular diseases, and traumatic brain injury. In this context, components of the endocannabinoid (eCB) system have been shown to shift microglia towards an anti-inflammatory activation state. Instead, much less is known about the functional role of the sphingosine kinase (SphK)/sphingosine-1-phosphate (S1P) system in microglia biology. In the present study, we addressed potential crosstalk of the eCB and the S1P systems in BV2 mouse microglia cells challenged with lipopolysaccharide (LPS). We show that URB597, the selective inhibitor of fatty acid amide hydrolase (FAAH)—the main degradative enzyme of the eCB anandamide—prevented LPS-induced production of tumor necrosis factor-α (TNFα) and interleukin-1β (IL-1β), and caused the accumulation of anandamide itself and eCB-like molecules such as oleic acid and *cis*-vaccenic acid ethanolamide, palmitoylethanolamide, and docosahexaenoyl ethanolamide. Furthermore, treatment with JWH133, a selective agonist of the eCB-binding cannabinoid 2 (CB_2_) receptor, mimicked the anti-inflammatory effects of URB597. Interestingly, LPS induced transcription of both SphK1 and SphK2, and the selective inhibitors of SphK1 (SLP7111228) and SphK2 (SLM6031434) strongly reduced LPS-induced TNFα and IL-1β production. Thus, the two SphKs were pro-inflammatory in BV2 cells in a non-redundant manner. Most importantly, the inhibition of FAAH by URB597, as well as the activation of CB_2_ by JWH133, prevented LPS-stimulated transcription of SphK1 and SphK2. These results present SphK1 and SphK2 at the intersection of pro-inflammatory LPS and anti-inflammatory eCB signaling, and suggest the further development of inhibitors of FAAH or SphKs for the treatment of neuroinflammatory diseases.

## 1. Introduction

Neuroinflammation generally refers to an inflammatory response of the innate immune system that occurs inside the brain or the spinal cord [1]. Microglia is one of the main mediators of the primary immune surveillance and phagocytic activity in the central nervous system (CNS) [2,3]. Resting microglia can detect soluble factors and switch into different phenotypes to maintain homeostasis: the M1-like phenotype is pro-inflammatory and marked principally by the release of the tumor necrosis factor-α (TNFα), interleukin (IL)-1β and other neuroinflammatory factors [4]. In contrast, the M2-like phenotype is anti-inflammatory and marked by interleukin (IL)-10 [3,5,6,7]. To sense their environment, microglia express pattern-recognition receptors (PRRs), including Toll-like receptors (TLRs) [8]. Therefore, they can recognize specific pathogen-associated molecular patterns (PAMPs), such as bacterial lipopolysaccharides (LPS) which are frequently found in bacterial pathogens [9].

However, in chronic neuroinflammation and neurodegeneration, pro-inflammatory microglia may contribute to detrimental progression [6,10]. Therefore, manipulation of microglia activation states might be a useful therapeutic approach. In the past few years, the endocannabinoid (eCB) system has come up as a regulator of microglia polarization [3,11,12]. The eCB system is composed of endogenous lipids termed endocannabinoids, such as anandamide (arachidonoylethanolamide, AEA) [13,14], which act as agonists at the G-protein-coupled cannabinoid 1 and 2 (CB_1_ and CB_2_) receptors, and also have other targets, e.g., the transient receptor potential vanilloid 1 (TRPV1) channel and the nuclear peroxisome proliferator-activated receptors (PPARs) α, δ and γ [15,16,17]. AEA is synthesized by the enzyme *N*-acyl phosphatidylethanolamine-specific phospholipase D (NAPE-PLD) and degraded by fatty acid amide hydrolase (FAAH) [18]. The eCB system is also composed of eCB-like compounds, such as palmitoylethanolamide (PEA) [19], oleic acid ethanolamide (OEA) together with its isomer *cis*-vaccenic acid ethanolamide (VEA) [20], which prolong eCBs activity by inhibiting their degradation with the so-called “entourage effect”. In addition, also ω-3 (n-3) fatty acid derivatives such as docosahexaenoyl ethanolamide (DHEA) belong to the eCB-like compounds [21]. Importantly, the majority of the receptors and enzymes of the eCB system are expressed in microglia [3,22]. In these cells, several studies have documented that eCB system activation promotes the anti-inflammatory M2-like phenotype and reduces neuroinflammation in animal models [3,12]. In particular, inhibition of FAAH with subsequent accumulation of various eCBs turned out to mediate protective effects [23,24]. In particular, the inhibition of FAAH activity by carbamoylation of the nucleophilic residue Ser241 through URB597, appeared successful [23,24].

Another bioactive lipid that plays a role in neuroinflammation and neurodegeneration is sphingosine-1-phosphate (S1P) [25,26]. S1P is generated through phosphorylation of sphingosine catalyzed by the sphingosine kinases 1 and 2 (SphK1 and 2), and acts as an agonist at five specific G-protein-coupled receptors, named S1P_1_-S1P_5_ [27,28]. Degradation of S1P is catalyzed by S1P lyase or occurs by recycling ceramides via diverse phosphatases and ceramide synthases [26,27]. Importantly, the G-protein-coupled S1P receptors have been established as useful targets for the treatment of the chronic neuroinflammatory disease, multiple sclerosis [29,30]. In particular, fingolimod, ponesimod, siponimod, and ozanimod have been approved for the treatment of relapsing-remitting and/or chronic progressive multiple sclerosis [30]. These drugs act mainly as super-agonists at the S1P_1_ receptor, inducing desensitization and internalization of the receptor, therefore disrupting the circulation of peripheral lymphocytes and preventing their invasion into the CNS [31]. In contrast to this well-established therapeutic mechanism, little information is available about the role of the SphK/S1P signaling system in the CNS-resident cells of the innate immune system, the microglia. A recent study suggests that the S1P_2_ receptor contributed to LPS-induced IL-6 secretion in mouse primary microglia and BV2 cells, since the effect of LPS was reduced by the S1P_2_ receptor antagonist, JTE-013 [32]. Two studies report that LPS, ischemia-reperfusion injury, or oxygen-glucose deprivation increased SphK1 expression in microglia, and suggested a pro-inflammatory role of this enzyme [33,34]. Considering the complex interplay of S1P receptors and SphKs in the related cell type, the macrophage [35], it is obvious that more work is required to address the functional role(s) of the S1P signaling system in microglia.

The present study aimed to investigate potential crosstalk between the eCB and the S1P systems in an inflammation model of murine BV2 microglia cells. Such crosstalk has been observed for example in skeletal muscle cells [36] and in the regulation of the vascular tone [37,38]. Here, we show that both SphK1 and SphK2 were induced by LPS and were required for pro-inflammatory cytokine production in BV2 cells, and that the FAAH inhibitor URB597, and CB_2_ receptor agonist reverted both LPS-induced cytokine production and SphK1/2 induction, thus placing SphKs at the intersection of pro-inflammatory LPS and anti-inflammatory eCBs signaling in these cells.

## 2. Results

### 2.1. FAAH Inhibition Reverts the LPS-Induced Pro-Inflammatory Response and SphK1/2 Induction

URB597 is a well-characterized [39] and widely used FAAH inhibitor that has been shown earlier to suppress pro-inflammatory responses in microglia [23,24]. Here, based on previous experiments [23,24,40] we treated BV2 microglia cells with 10 µM URB597 for 30 min before challenging them with 100 ng/mL LPS for 4 h, with or without the continuous presence of URB597. Of note, URB597 at concentrations of 0.01–10 µM, or 10 µM URB597 in combination with LPS, did not show any cytotoxicity in an MTT assay (Appendix A). As shown in Figure 1a, treatment with LPS caused significant induction of TNFα and IL-1β mRNA, while URB597 efficiently counteracted the effect of LPS. In addition, URB597 increased the expression of anti-inflammatory IL-10 (Appendix A). Furthermore, we show that mRNA levels of both SphK1 and SphK2 increased after 4 h of treatment with 100 ng/mL LPS (Figure 1b). Most importantly, URB597 fully prevented LPS-induced induction of both SphK1 and SphK2 (Figure 1b). 

Sphingolipid measurements by LC-MS/MS revealed that levels of S1P d18:1 after 4 h of incubation with LPS were not altered in BV2 cells at this relatively early time point (97.7 ± 9.3% of control; means ± SEM; *n* = 4), in agreement with a report from macrophages [41]. However, levels of sphingosine d18:1 were slightly but significantly decreased (91.3 ± 2.3% of control; means ± SEM; *n* = 4; *p* < 0.05). This observation suggests a rapid turnover of S1P that was formed by the up-regulated SphKs, leading to the consumption of the substrate. Interestingly, treatment with URB597 for 4 h in the absence of LPS significantly decreased cellular levels of S1P d18:1 to 78.7 ± 5.7% of control (means ± SEM; *n* = 4; *p* < 0.05).

Since the functional roles of the two SphKs are strongly dependent on their subcellular localization [42], we studied the localization of SphK1 and SphK2 in treated and untreated BV2 cells by immunofluorescence and confocal laser scanning microscopy. Usually, SphK1 is mainly a cytosolic enzyme that can translocate to the plasma membrane upon activation, while SphK2 may be localized in the cytoplasm, nucleus, mitochondria, or endoplasmic reticulum [43]. Surprisingly, our results showed that not only SphK2 but also SphK1 was predominantly localized in the nucleus of BV2 cells (Figure 2a,b). There were only a few cells that showed little nuclear staining of SphK1. Some SphK1 immunoreactivity was found in the cytosol, while there was very little cytosolic SphK2 (Figure 2a,b). The predominantly nuclear localization of SphK1 in BV2 cells was observed with antibodies from both Proteintech (Figure 2) and Santa Cruz Biotechnology. FAAH, on the other hand, was observed at dots and membranous structures within BV2 cells (Figure 2c), in agreement with its structure as a transmembrane protein [44]. Importantly, none of the enzymes changed their location after treatment with LPS and/or URB597 (Figure 2a–c).

### 2.2. Involvement of SphK1 and Sphk2 in the Pro-Inflammatory Response to LPS

SphK1, SphK2, and S1P may act as pro- or anti-inflammatory mediators, depending on the cellular system and conditions [32,35]. Here, we studied the role of the two SphK isoenzymes in LPS-induced inflammation in BV2 cells using the highly selective and effective SphK inhibitors, SLP7111228 for SphK1 [45], and SLM6031434 for SphK2 [46]. mRNA levels of the pro-inflammatory cytokines TNFα and IL-1β were evaluated after pretreatment with 1 μM SLP7111228 and/or 1 µM SLM6031434 for 30 min before the addition of 100 ng/mL LPS for 4 h. Importantly, both SphK inhibitors strongly and significantly reverted the effect of LPS on IL-1β mRNA expression (Figure 3a,b). Furthermore, SLM6031434 significantly decreased mRNA expression of IL-1β, while the reduction of TNFα mRNA by SLP7111228 was not significant, due to high variability between experiments (Figure 3a). The two inhibitors used together had the same effect as either inhibitor used alone on both TNFα and IL-1β in inflamed cells (Figure 3a,b).

### 2.3. CB_2_ Agonism Reduces Inflammation and SphKs Expression after LPS Induction

Since FAAH inhibition determines an increase in eCBs levels, we first analyzed to what extent specific eCBs contents were affected by URB597 with and without LPS in BV2 cells. As shown in Figure 4, upon 30 min pretreatment with 10 μM URB597, both in the absence and presence of LPS, the most abundant eCBs were OEA together with its isomer VEA, and PEA (Figure 4b,c). These eCB-like compounds increased by ~6-fold and ~2-fold, respectively. Moreover, DHEA, which was ~10-fold less abundant that the three eCB-like molecules, increased by ~5-fold, Figure 4d. AEA levels were below the lower limits of quantification (LLOQ) in control cells and cells treated with LPS, but it became measurable and increased more than 3-fold (compared to the LLOQ) in cells treated with URB597, or URB597 plus LPS (Figure 4a).

The increase in the content of AEA and congeners led us to interrogate the role of eCB-binding receptors, first CB_1_ and CB_2_, due to their well-known involvement [47,48]. Therefore, the protein expression of these two receptors was analyzed in BV2 cells treated with 100 ng/mL LPS and 10 μM URB597 for 4 h, by Western blotting. Results showed that the CB_1_ receptor was poorly expressed but appeared significantly increased during treatment with URB597 together with LPS (Figure 5a). Instead, the CB_2_ receptor was highly expressed with a significant increase induced by URB597 treatment, with and without LPS (Figure 5b). The CB_2_ / CB_1_ protein expression ratio in control and LPS-treated cells CB_2_ is at least 7 times higher than CB_1_ (7.69 ± 0.15 CTRL; 7.85 ± 0.09 LPS). This ratio was shifted even more towards the CB_2_ receptor upon treatment with URB597 (11.10 ± 0.20), in keeping with CB_2_ up-regulation by URB597 shown in Figure 5. In URB597 treatment with LPS, the ratio went lower than in the control (6.42 ± 0.23).

Finally, we addressed the question of whether the effects of the FAAH inhibitor URB597 on pro-inflammatory markers and SphK1/2 expression could be mimicked by a CB_2_ agonist. To this end, we used the highly selective CB_2_ agonist, JWH133 [49]. Of note, neither 0.01–10 µM JWH133 nor 1 µM JWH133 together with LPS showed cytotoxicity in an MTT cytotoxicity assay (Appendix A). As shown in Figure 6a,b, pretreatment for 30 min with 1 µM of JWH133 significantly reverted the LPS-induced increase of TNFα and IL-1β mRNA [50,51,52]. Furthermore, similar to URB597, also JWH133 increased the level of anti-inflammatory IL-10 (Appendix A). These results suggest that CB_2_, in agreement with its previously reported anti-inflammatory role [47,48], is involved in the anti-inflammatory eCBs signaling after treatment with URB597. Most importantly, the selective CB_2_ agonist fully reverted the induction of SphK1 and SphK2 by LPS (Figure 6c,d).

Altogether, these results demonstrate that SphKs are a point of intersection between the pro-inflammatory (LPS) and anti-inflammatory (eCBs/CB_2_) signaling pathways.

## 3. Discussion

Accumulated evidence has associated the pathophysiology of neuroinflammatory and neurodegenerative diseases with pro-inflammatory microglia [6,10]. These resident immune cells of the brain are fundamental to maintaining CNS homeostasis and protecting against harmful agents, but may also drive deleterious mechanisms in chronic neuroinflammation and neurodegeneration [6,10]. In response to the microenvironment, microglia adopt specialized phenotypes, described as polarization, and may produce preferentially either pro- or anti-inflammatory mediators [6,24,53].

Due to the unprecedented evidence of the influence of the eCB system on the S1P system, and vice versa, in cultured murine skeletal-muscle C2C12 cells [36], in the regulation of rat coronary artery reactivity [37] and in the mediation of the blood pressure changes in mice [38], in this study the possible functional interplay between the S1P and the eCB systems has been explored in inflamed murine microglial BV2 cells.

In the past few years, the eCB system has been identified as one of the key endogenous regulators of microglia polarization [11,53]. Indeed, several studies have demonstrated that its activation can inhibit cytokine and chemokine production, thus polarizing microglia towards the anti-inflammatory (M2) state [11,53,54]. One possibility for pharmacological activation of the eCB system is the inhibition of the eCB-degrading enzyme, FAAH [44]. Here, we used LPS-treated BV2 cells as a model for pro-inflammatory microglia and demonstrated that induction of the pro-inflammatory markers, TNFα and IL-1β, was fully prevented by the prior addition of the FAAH inhibitor, URB597. Our data thus validate the efficacy of FAAH inhibition in reverting an inflamed state in microglia [23,24,40]. 

Moreover, we demonstrated that in BV2 microglia FAAH inhibition modulated principally AEA and its eCB-like congeners OEA, VEA, PEA, and DHEA, which are already known to ameliorate the neuroinflammatory status [22]. In particular, AEA was found to be released by M2-polarized microglia cells and to stimulate CB_1_ and CB_2_ expression [55]. OEA and PEA were found to be neuroprotective in a murine model of LPS-induced inflammation, where they were able to reduce IL-1β, TNF-α, cyclooxygenase-2, and prostaglandin E_2_ [56]. Moreover, in N9 microglia cells and primary microglia PEA was found to prevent Ca^2+^ transients and neuronal hyperexcitability induced by LPS, through the interaction with a CB_2_-like receptor [57]. Finally, in an LPS-induced BV2 model, a DHEA-derived synaptamide was able to reduce inflammation through the cAMP/protein kinase A pathway and the inhibition of the nuclear factor kappa-light-chain-enhancer of activated B cells (NF-κB) activation [58].

Here, the increase of anti-inflammatory OEA/VEA, PEA, and DHEA upon URB597 treatment supports the validity of FAAH inhibition as a strategy to counteract acute neuroinflammatory development [23,24,40]. On the other hand, the anti-inflammatory effect of eCB-like compounds may also be due to an “entourage effect” that inhibits FAAH activity and mimics URB597 action, thus extending the lifetime of true eCBs such as AEA [59,60,61].

Furthermore, the major eCB-binding CB_2_ receptor is known to regulate microglia towards an anti-inflammatory phenotype, both in vitro [46,47,62,63] and in vivo in a mouse model of Alzheimer’s disease [64]. Our data confirm that CB_2_ is the predominant cannabinoid receptor in BV2 microglial cells, and that its activation by the agonist JWH133 efficiently reduces LPS-induced production of TNFα and IL-1β and thus, the inflammatory state. Interestingly, CB_2_ protein expression increased upon treatment with URB597, consistent with its activation by the accumulated eCBs, no longer cleaved by an inhibited FAAH. In this context, also additional receptors may be engaged by eCBs in the inflammatory response, as well as by eCB-like compounds that do not bind to CB_2_ (nor CB_1_) receptors, and yet can act via TRPV1 and PPAR-α [65]. Regarding TRPV1, its influence on microglia has been well-established in recent years [3]. For instance, it can regulate NLRP3 inflammasome activation and consequently mediate neuroinflammation [66]. Therefore, the increased eCB levels observed upon FAAH inhibition may be protective also by engaging TRPV1 and thus reducing inflammasome activation. Future research efforts will be required to clarify this point.

The role of the S1P signaling system in neuroinflammation is best illustrated by the success of the S1P receptor modulators such as fingolimod, siponimod, and ponesimod in the treatment of multiple sclerosis [30]. These drugs act mainly by internalizing the S1P_1_ receptor, which is required for lymphocyte circulation, thus preventing the entry of auto-reactive peripheral lymphocytes into the brain [31]. However, surprisingly little is known about the role of the SphK/S1P signaling system in the innate immune cells of the brain, the microglia [26,67]. Nayak and colleagues have shown that LPS-induced SphK1 expression in BV2 cells, and suggested a pro-inflammatory role of this kinase because LPS-induced release of pro-inflammatory cytokines was inhibited by *N*,*N*-dimethylsphingosine [33]. The latter substance, however, is notorious for its non-specific effects [68]. SphK1 was induced in microglia after cerebral ischemia [69], and the knockdown of SphK1 decreased IL17A production after oxygen/glucose deprivation [34]. More recent work by Karunakaran and colleagues has shown that LPS-induced IL-6 production in primary mouse microglia and BV2 cells was inhibited by the S1P_2_ receptor antagonist, JTE-013, therefore supporting a pro-inflammatory role of extracellular S1P and the S1P_2_ receptor in microglia [32]. In the microglia-related cell type, the macrophage, detailed work has shown the complexity of S1P functions and the distinct engagement of the different S1P receptors [35]. Of note, the role of the SphKs in macrophages remains controversial, as genetic deletion of SphK1, SphK2, or both had no major influence on LPS-induced cytokine expression [70,71]. Other data, however, showed a pro-inflammatory role for SphK1 [72], while SphK2 appeared to act as a negative regulator of inflammatory macrophage activation [73]. 

Here, we focused on the LPS-mediated regulation of SphKs in BV2 microglia cells. We show that transcription of both SphK1 and SphK2 was induced by LPS, and that both kinases were required for LPS-induced expression of TNFα and IL-1β mRNA. S1P levels were not increased after 4 h of incubation with LPS, in agreement with earlier results showing an increase in S1P only after 16 h of LPS treatment [41]. However, the small but significant consumption of sphingosine suggests that a localized production and rapid metabolism of S1P occurred after SphK1/2 induction. We used the novel, highly specific SphK inhibitors, SLP7111228 and SLM6031434, and therefore circumvented counter-regulatory cellular mechanisms which may occur in SphK knockout models [46,70]. Interestingly, inhibition of a single SphK isoform was as efficient as inhibition of both SphK1/2, showing that SphK1 and SphK2 were not redundant in BV2 cells and that both enzymes were required. Regarding the mechanisms by which the SphKs contributed to LPS-induced pro-inflammatory responses, it is important that both enzymes were mainly localized in the nucleus of BV2 cells. Although nuclear SphK2 is common and might, for example, have epigenetic effects [43], SphK1 has rarely been observed in the nucleus. Nevertheless, it should be noted that not only SphK2 but also SphK1 has nuclear export sequences, and their deletion trapped the enzyme in the nucleus [74,75]. Therefore, in microglia, SphK1 probably has specific effects within the nucleus. The comparably minor part of SphK1 in the cytosol of the cells, shown in Figure 2, might however enable autocrine S1P secretion and activation of S1P receptors after exposure to LPS, as suggested by Karunakaran and colleagues [32]. Future work is deemed necessary to address the detailed mechanisms by which SphK1 and SphK2 act as pro-inflammatory agents in microglia cells.

The most important observation in the present study is that both FAAH inhibition and CB_2_ agonism reduced not only pro-inflammatory cytokine production but also suppressed LPS-induced transcription of both SphK1 and SphK2. The effect of the FAAH inhibitor is likely due to the accumulation of eCBs that act at the CB_2_. Moreover, to the best of our knowledge, this is the first report on the inhibition of SphK1 and SphK2 transcription by a G_i_-coupled receptor, while rapid activation and membrane translocation [76], as well as transcriptional induction [77] of SphK1 by Gq-coupled receptors, have been already described. Of note, no change in the location of either SphK1 or SphK2 was observed after incubation with the FAAH inhibitor URB597. This observation is in line with a previous finding showing that G_i_-coupled receptors did not induce translocation of (cytosolic) SphK1 [75]. Since SphK1 transcription is induced for example via mitogen-activated protein kinases and AP1 transcription factor, it might be speculated that the CB_2_ receptor, counteracted this induction by activating a mitogen-activated protein kinase phosphatase as indeed shown in a previous study [48]. Other potential mechanisms for the down-regulation of SphK mRNA might involve micro-RNAs, since they are both targets of eCB signaling and effectors of SphK transcription [78,79]. Again, future studies must be performed to address this unprecedented mechanism of G_i_-mediated SphK1/2 suppression.

## 4. Materials and Methods

### 4.1. Materials and Reagents

Dulbecco’s modified Eagle’s medium with high glucose content and GlutaMAX, fetal calf serum, and penicillin/streptomycin were from Gibco-Thermo Fisher Scientific (Waltham, MA, USA). Lipopolysaccharide (LPS) from E. coli (O111:B4, #L2630), URB597 (#U4133), and JWH133 (#10005428) were from Cayman Chemical (Ann Arbor, MI, USA). The SphK1 and 2 inhibitors, SLP7111228 (#857380P) and SLM6031434 (#857381P) were from Avanti Polar Lipids (Alabaster, AL, USA).

### 4.2. Cell Culture and Treatment

BV2 microglia cells were cultured in Dulbecco’s modified Eagle’s medium (DMEM) with GlutaMAX, supplemented with 10% fetal calf serum and 100 U/mL penicillin/streptomycin at 37 °C in a humidified 5% CO_2_ atmosphere. When 90% confluent, cells were pretreated for 30 min with 10 μM URB597, 1 μM JWH133, or 1 µM each of SLP7111228 and/or SLM6031434, before exposure to 100 ng/mL LPS for 4 h, as reported [23,75]. All the cited compounds are dissolved in DMSO, thus cells stimulated with DMSO (≤0.1%) as vehicle were used as control. 

### 4.3. MTT Viability Assay

Cells were seeded into 96-well plates two days before treatment, at a density of 1.5 × 10^4^ cells. After 48 h, cells were incubated with URB597 or JWH133 at different concentrations (0.01, 0.1, 1.0, 5.0, 10 μM), and together with LPS at the concentrations chosen for the other experiments (10 μM for URB597 and 1 μM for JWH133), for 4 h. Then, cell viability was assessed by the mitochondrial-dependent reduction of 3-[4,5-dimethylthiazol-2-yl]-2,5-diphenyl tetrazolium bromide (MTT; Sigma, St. Louis, MO, USA) to purple formazan. After 3 h of incubation, the MTT was discarded and 150 μL of DMSO was added to dissolve the formazan crystals. Absorbance was detected by Enspire multimode plate reader (Perkin Elmer, Waltham, MA, USA). The cell viability was calculated by subtracting the 630 nm OD background from the 570 nm OD total signal of the cell-free blank of each sample and was expressed as a percentage of controls set to 100% [36].

### 4.4. Quantitative Real Time-Reverse Transcriptase-Polymerase Chain Reaction (qRT-PCR)

mRNA was isolated using a standardized phenol-based method, according to the modified method by Chomczynski and Sacchi [80]. It was quantified using a NanoDrop spectrophotometer (Thermo Fisher Scientific) and transcribed into cDNA using a RevertAid First Strand cDNA Synthesis Kit (Applied Biosystems/Thermo Fisher Scientific) according to the manufacturer’s instructions. qRT-PCR was performed with the Applied Biosystems 7500 Fast Real-Time PCR System. The following TaqMan probes were used: 18S (Mm03928990_g1), TNFα (Mm00443258_m1), IL-1β (Mm00434228_m1), SphK1 (Mm00448841_g1), SphK2 (Mm00445021_m1), from Applied Biosystems (Waltham, MA; USA). SensiFAST^TM^ SYBR Lo-ROX kit was used to assess the relative abundance of the following genes in Figure 3 and Appendix A: β-actin (Forward 5′ → 3′ TGTTACCAACTGGGACGA; Reverse 5′ → 3′ GTCTCAAACATGATCTGGGTC); GAPDH (Forward 5′ → 3′ AACGGGAAGCTCACTGGCAT; Reverse 5′ → 3′ GCTTCACCACCTTCTTGATG); TNF-α (Forward 5′ → 3′ GATCGGTCCCCAAAGGGATG; Reverse 5′ → 3′ GTTTGCTACGACGTGGGCT); IL-1β (C ACGGACCCCAAAAGATGAAGGGCT; Reverse 5′ → 3′ GGGAACGTCACACACCAGCAGG) and IL-10 (Forward 5′ → 3′ GCACTACCAAAGCCACAAAGC; Reverse 5′ → 3′ GTCAGTAAGAGCAGGCA).

The relative expression of different amplicons was calculated by the 2^−ΔΔCt^ method, using 18S (Figure 1 and Figure 6) or the mean of β-actin and GAPDH (Figure 3 and Appendix A) as reference genes [81].

### 4.5. Western Blotting

Treated cells were lysed in RIPA buffer in the presence of a protease inhibitors cocktail (Sigma-Aldrich, St. Louis, MO, USA). Then, they were sonicated three times and centrifuged at 9500× *g* for 15 min at 4 °C; supernatants were collected, and protein content was determined by the Bio-Rad Protein assay (Bio-Rad Laboratories, Hemel Hempstead, UK). Cell lysates were mixed with Laemmli sample buffer (heated for 10 min at 60 °C) so that an equal amount of protein per lane (60 μg) was subjected to a 10% sodium dodecyl sulfate–polyacrylamide gel electrophoresis (SDS-PAGE). Gels were then electroblotted onto polyvinylidene difluoride (PVDF) membranes (Amersham Hybond, GE Healthcare Life Science, Piscataway, NJ, USA). Subsequently, PVDF membranes were blocked with 5 % milk, incubated overnight with the primary antibodies at 4 °C, and then with the appropriate horseradish peroxidase-conjugated secondary antibody (#31461, Thermo Fisher Scientific, Waltham, MA, USA) for 1 h at room temperature. Primary antibodies specific for the following proteins were used: β-actin (#SAB5600204, Sigma-Aldrich, St. Louis, MO, USA), CB1 C-Terminal (#10006590, Cayman Chemical, Ann Arbor, MI, USA), CB2 (#101550, Cayman Chemical). Detection was performed by ClarityWestern ECL substrate (Bio-Rad, Hercules, CA, USA) as developed by Azure Biosystems c400 (Sierra Ct, Dublin, CA, USA). Immunoreactive band intensities were quantified by densitometric analysis through the ImageJ software v.1.53K (NIH, Bethesda, MD, USA).

### 4.6. Immunocytochemistry and Fluorescence Microscopy

BV2 cells were fixed with 4% paraformaldehyde for 1 h on ice, and permeabilized with 0.1% Triton X-100 in PBS for 5 min at room temperature. Then, they were blocked with 5% milk at room temperature for 1 h. Next, they were incubated with primary antibodies for FAAH (#101600, 1:100, Cayman Chemical, Ann Arbor, MI, USA), SphK1 (#10670-1-AP, 1:100, Proteintech, Manchester, UK; #Sc-365401, 1:100, Santa Cruz Biotechnology, Heidelberg, Germany) or SphK2 (#ab37977, 1:50, Abcam, Cambridge, UK) for 1 h at room temperature. Thereafter, an incubation with Alexa Fluor 488-conjugated secondary antibodies (1:1000) was performed for 1 h in the dark at room temperature. Finally, the cells were washed with PBS and stained with DAPI (1 µg/mL) for 90 s in the dark at room temperature. Confocal laser scanning microscopy was performed with a Zeiss LSM510 Meta system equipped with an inverted Observer Z1 microscope and a Plan-Apochromat 63×/1.4 oil immersion objective (Carl Zeiss MicroImaging GmbH, Jena, Germany). The following excitation (ex) laser lines and emission (em) filter sets were used: DAPI: ex 405 nm, em band-pass 420–480 nm; Alexa Fluor 488: ex 488 nm, em long-pass 505 nm. Spatial calibration of the images was carried out with the “set scale” function. Microscopic images were processed with the ZEN software v.2009 (Carl Zeiss MicroImaging GmbH).

### 4.7. Liquid Chromatography-Tandem Mass Spectrometry

The quantification of sphingolipids and endocannabinoids was performed by liquid chromatography-tandem mass spectrometry (LC-MS/MS). Sphingolipids were determined as described for the analysis of cell pellets in detail before [82]. The analysis of endocannabinoids was performed by adapting a method that was originally developed for the analysis of plasma samples. Briefly, cell pellets were resuspended in 200 µL of water and extracted using liquid-liquid-extraction as previously described in detail [83].

### 4.8. Statistical Analysis

Data were analyzed by the GraphPad Prim program version 9 (GraphPad Software, La Jolla, CA, USA), and reported as means ± SD of experiments performed in triplicates (Figure 4), or as means ± SEM of the indicated number of independent experiments performed in triplicates (all other experiments). Statistical analysis was performed as indicated in the figure legends. *p* < 0.05 was considered statistically significant.

## 5. Conclusions

In conclusion, the present study provides unprecedented evidence that the anti-inflammatory effects of FAAH inhibition in LPS-stimulated microglia involve the suppression of pro-inflammatory SphK1 and SphK2 transcription. Furthermore, activation of the CB_2_ receptor directly counteracted the induction of SphK1 and SphK2, showing an unanticipated suppression of SphK transcription by a G_i_-coupled receptor. These novel findings are summarized in Figure 7. Further studies are deemed necessary to address the mechanistic details of pro-inflammatory signaling by nuclear SphKs, as well as the pathways whereby SphK1/2 induction is counteracted by the CB_2_ receptor. Finally, it seems relevant to evaluate in an independent investigation whether ω-3 eCBs and eCB-like compounds may indirectly influence CB_2_ activity via an “entourage effect”, and/or directly by binding to TRPV1 and PPARα receptors.

## Figures and Tables

**Figure 1 ijms-24-08508-f001:**
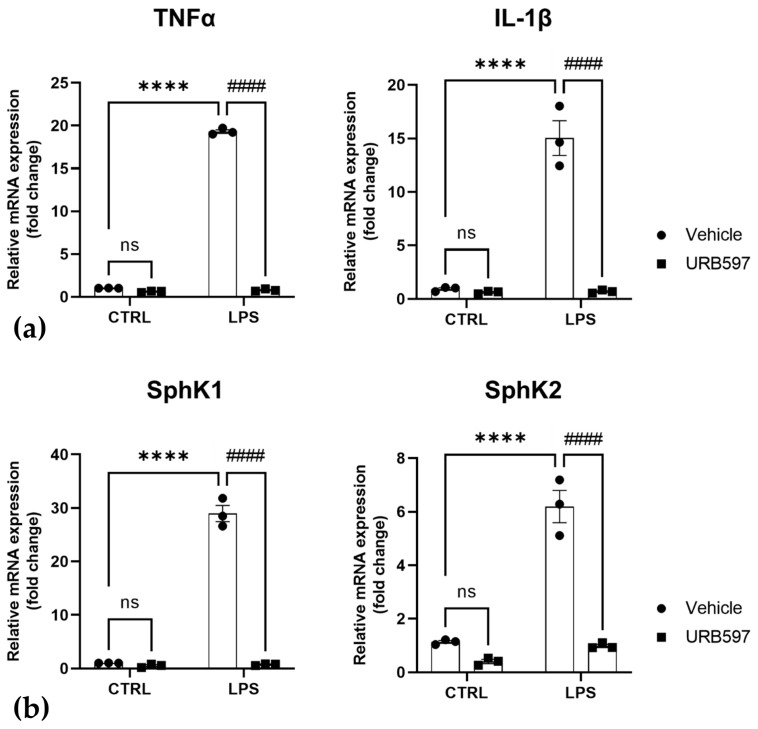
Inhibition of LPS-induced inflammation (**a**) and SphK1/2 induction (**b**) by the FAAH inhibitor, URB597. BV2 cells were pretreated with vehicle (CTRL) or 10 µM URB597 for 30 min, and then further incubated with vehicle or 100 ng/mL LPS for 4 h. mRNA levels of TNFα, IL-1β, SphK1, and SphK2 were determined by qRT-PCR and expressed as fold change, using the 2-ΔΔCT method. Data are means ± SEM of three independent experiments each in (**a**,**b**). Statistical analysis was performed by two-way ANOVA followed by Tukey’s multiple comparisons test (**** *p* < 0.0001 vs. CTRL; #### *p* < 0.0001 vs. LPS; ns, not significant).

**Figure 2 ijms-24-08508-f002:**
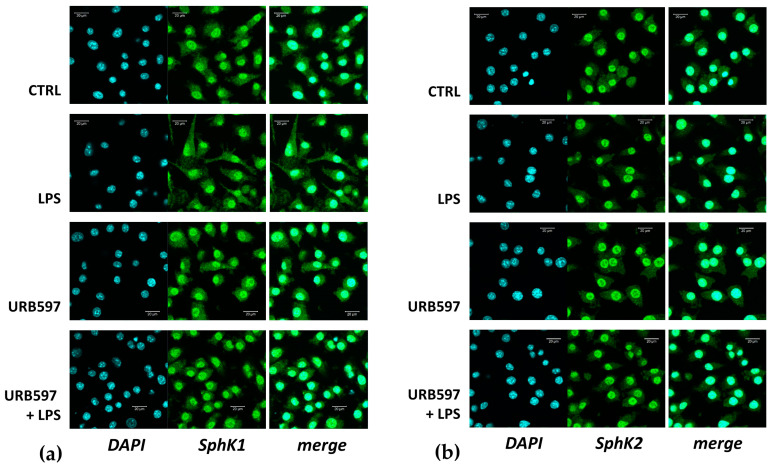
Location of SphK1, SphK2, and FAAH in BV2 cells treated with or without LPS and/or URB597. BV2 cells were pretreated with vehicle or 10 µM URB597 for 30 min, and then incubated with vehicle (CTRL) or 100 ng/mL LPS for 4 h. Localization of SphK1 (**a**), SphK2 (**b**), and FAAH (**c**) was analyzed by immunocytochemistry and confocal laser scanning microscopy. Shown are representative images. Bars: (**a**) 20 µm; (**b**) 20 µm; (**c**) 10 µm.

**Figure 3 ijms-24-08508-f003:**
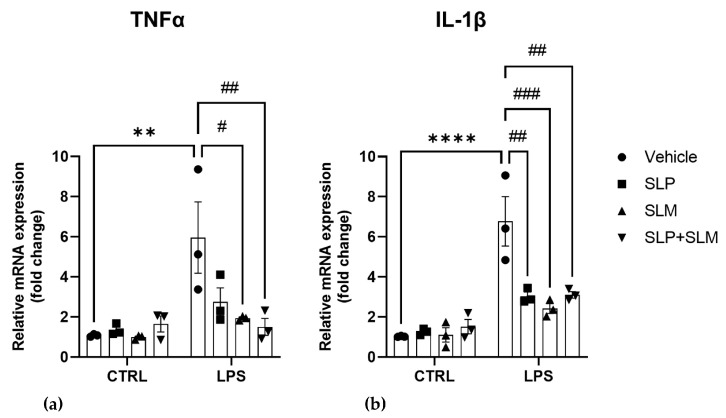
Pro-inflammatory role of SphKs in LPS-treated BV2 cells. BV2 cells were pretreated with vehicle (CTRL), (**a**) 1 µM of the SphK1 inhibitor SLP7111228 (SLP), (**b**) 1 µM of the SphK2 inhibitor SLM6031434 (SLM), or both SLP7111228 and SLM6031434 for 30 min, and then further incubated with vehicle or 100 ng/mL LPS for 4 h. mRNA levels of TNFα and IL-1β were measured by qRT-PCR and expressed as fold change, using the 2^−ΔΔCT^ method. Data as means ± SEM of three independent experiments. Statistical analysis was performed by two-way ANOVA followed by Tukey’s multiple comparisons test (** *p* < 0.01, **** *p* < 0.0001 vs. CTRL; # *p* < 0.05, ## *p* < 0.01, ### *p* < 0.001 vs. LPS).

**Figure 4 ijms-24-08508-f004:**
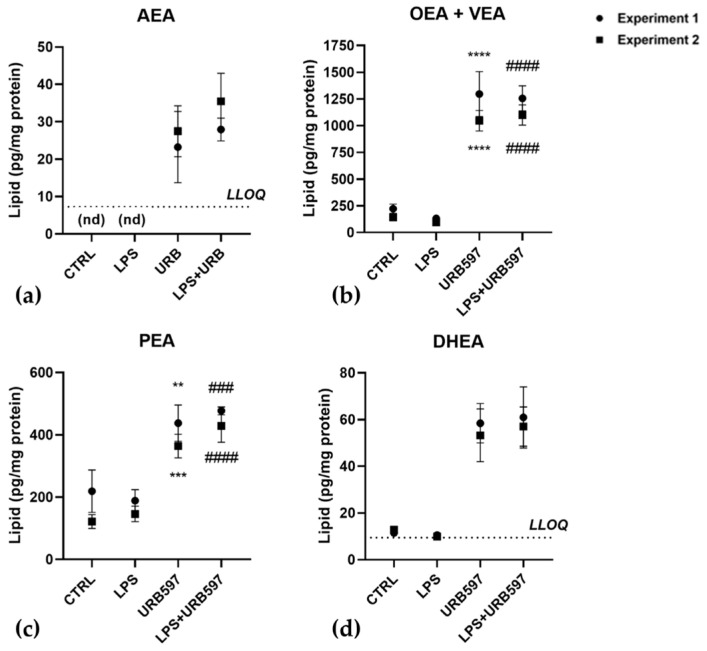
Effects of LPS and URB597 on levels of endocannabinoids in BV2 cells. BV2 cells were pretreated with vehicle or 10 µM URB597 for 30 min, and then incubated with vehicle (CTRL) or 100 ng/mL LPS for 4 h. Levels of endocannabinoids were measured by LC-MS/MS. (**a**) AEA, anandamide; (**b**) OEA, oleic acid ethanolamide; VEA, cis-vaccenic acid ethanolamide; (**c**) PEA, palmitoylethanolamide; (**d**) DHEA, docosahexaenoyl ethanolamide. Values are means ± SD of two independent experiments, each performed in triplicate. The dotted lines show the lower limits of quantification (LLOQ). AEA levels in CTRL and LPS-treated cells were below the LLOQ and are designated and not detectable. Statistical evaluation was performed separately for the two experiments using two-way ANOVA followed by Tukey’s multiple comparisons test (** *p* < 0.01, *** *p* < 0.001, **** *p* < 0.0001 vs. CTRL; ### *p* < 0.001, #### *p* < 0.0001 vs. LPS).

**Figure 5 ijms-24-08508-f005:**
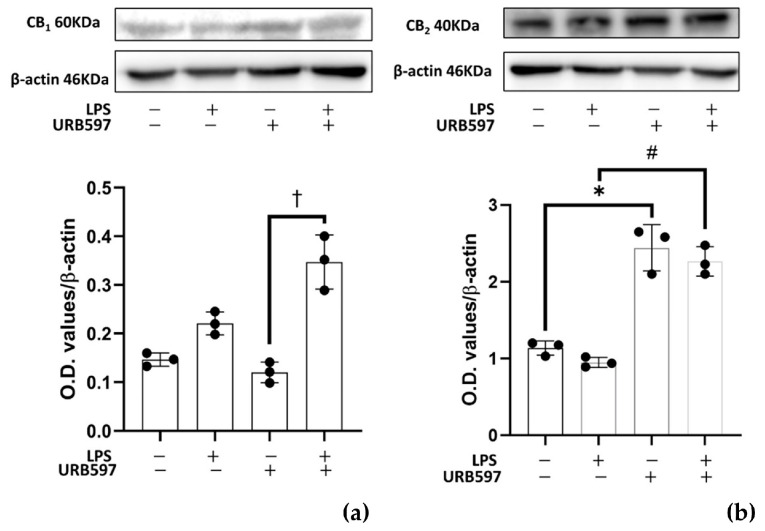
Effects of LPS and URB597 on CB_1_ (**a**) and CB_2_ (**b**) protein expression. BV2 cells were pretreated with vehicle or 10 µM URB597 for 30 min, and then incubated with vehicle or 100 ng/mL LPS for 4 h. Protein levels were normalized to β-actin. Data are means ± SEM of three independent experiments. Statistical analysis was performed by one-way ANOVA followed by Dunnett’s multiple comparisons test. (* *p* < 0.05 vs. CTRL; # *p* < 0,05 vs. LPS; † *p* < 0,05 vs. URB597).

**Figure 6 ijms-24-08508-f006:**
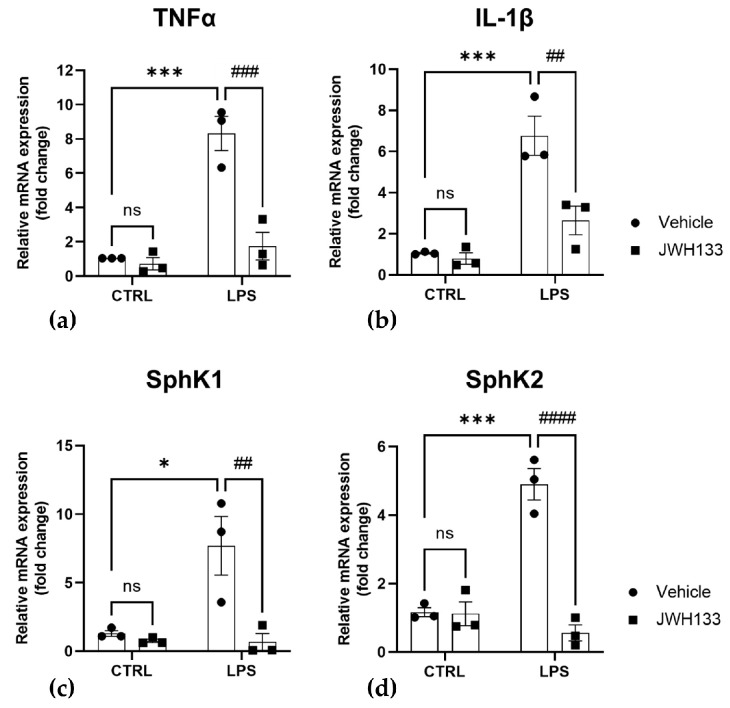
Reversal of LPS-induced inflammation (**a**,**b**) and SphK1/2 induction (**c**,**d**) by CB_2_ receptor agonist. BV2 cells were pretreated with vehicle (CTRL) or 1 µM of JWH133 for 30 min, and then incubated with vehicle or 100 ng/mL LPS for 4 h. mRNA levels of TNFα, IL-1β, SphK1, and SphK2 were determined by qRT-PCR and expressed as fold change, using the 2^−ΔΔCT^ method. Data are means ± SEM of three independent experiments. Statistical analysis was performed by two-way ANOVA followed by Tukey’s multiple comparisons test (* *p* < 0.05, *** *p* < 0.001 vs. CTRL; ## *p* < 0.01, ### *p* < 0.001, #### *p* < 0.0001 vs. LPS).

**Figure 7 ijms-24-08508-f007:**
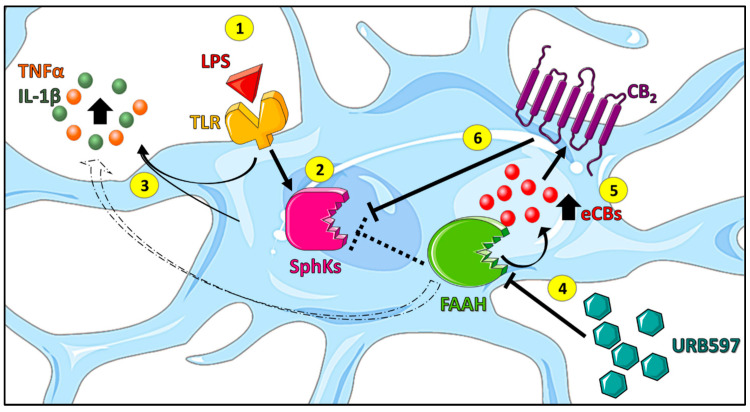
Summary of the main findings. In BV2 microglia, the binding of LPS to TLRs (1) leads to an increased expression of SphKs (2), which then contributes to LPS-induced induction of the pro-inflammatory cytokines, TNFα and IL-1β (3). The use of URB597 (FAAH inhibitor) leads to an increase in eCBs (4), which, in turn, can activate the CB_2_ receptor (5) and thus reduce inflammation (i.e., pro-inflammatory cytokine release) as well as SphK levels (6). The Figure was partly generated using Servier Medical Art, provided by Servier (Suresnes, France), licensed under a Creative Commons Attribution 3.0 unported license.

## Data Availability

All experimental data are presented in the article.

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
