# Peer review of "Sphingosine Kinases at the Intersection of Pro-Inflammatory LPS and Anti-Inflammatory Endocannabinoid Signaling in BV2 Mouse Microglia Cells"

_ijms, 2023, doi:10.3390/ijms24108508_

Round 1
Reviewer 1 Report
Comments to the Author
Comments for ijms-2310187.
Manuscript entitled “Sphingosine kinases at the intersection of pro-inflammatory 2 LPS and anti-inflammatory endocannabinoid signalling in BV2 mouse microglia cells” by Standoli and colleagues. This study investigates the crosstalk of the eCB and the S1P systems in BV2 mouse microglia cells challenged with LPS. I found the subject of some interest, however, I have several issues on the methodology and therefore on data interpretation to be addressed before publication.
See below for some specific comments.
Major issue:
1. A significant limitation of this study is that the authors use a pretreatment protocol that is not applicable to the clinic. The authors should explain the choice of the protocol used for the study.
2. A limitation of this study is that the authors use only one dose of URB 597 and JWH133 for the in vitro experiments. The authors should explain the choice of the dose used for the study.
3. The authors considered URB597 as a selective inhibitor of FAAH. However, for this latter mechanism, URB597 showed a maximal activity at 1 μM (Tanaka et al., 2019), the authors used the drug at much higher concentrations. The authors should discuss this discrepancy.
4. The aim of this study looks unfocused. In particular, the use of URB597 induced an accumulation of anandamide, that is an endocannabinoid and also an endovanilloid. A better discussion of the possible involvement of CB1 and TRPV1 receptors on the crosstalk of the eCB and the S1P systems is needed.
Minor issue:
1. Please change the number of figure on immune image (Fig. 2 vs Fig. 1). Also please explain why do you use different scale bar.
2. Please comment in Fig. 3 why SLP is not able to reduce significantly the increased mRNA levels of TNFα induced by LPS treatment.
3. Please change the WB image of CB1.
Author Response
Responses to Reviewer 1
Major issue:
A significant limitation of this study is that the authors use a pretreatment protocol that is not applicable to the clinic. The authors should explain the choice of the protocol used for the study.
R: Thanks for pointing this out. We agree with this comment, therefore we have improved the Materials and Methods section by including a paragraph (page 11-12, lines 373-377) on the protocol used to set up the model of inflamed BV2 cell in the revised manuscript.
Please also note that our study does not aim at identifying therapeutics, thus the clinical application of our finding appears out of scope; the experimental approach is aimed at evaluating only the “in vitro” effects of ECS modulation (FAAH inhibition and CB2 activation) on SphK1/2 expression in a cellular model of inflamed microglia.
A limitation of this study is that the authors use only one dose of URB597 and JWH133 for the in vitro experiments. The authors should explain the choice of the dose used for the study.
R: The dose of URB597 used in our experimental model (10 µM) is in keeping with the studies by Tham et al., 2007 - FEBS Lett. (this new reference has been included in the revised manuscript, ref.40) and by Tanaka et al. 2019 (ref. 23), where it seemed the most effective to block both FAAH activity and the inflammatory state of microglia. However, we have performed additional dose-dependence (0.01, 0.1, 1.0, 5.0 and 10 µM) studies with URB597 by confirming no-cytotoxicity also when BV2 cells were treated with LPS in combination with URB597 at the maximum dose of 10 µM. In addition, the effect of URB597 on anti-inflammatory IL-10 gene expression, by using qRT-PCR, has been tested with a new experiment. Therefore, these new results support the choice of 10 µM as the optimal time to produce significant effects. These new results are now included in Supplementary materials.
As for JWH133, we used the dose of 1 µM to selectively activate CB2 because this is the non-toxic effective dose that has been largely used in previous “in vitro” experiments on different cells (Compagnucci et al., Plos One 2013, ref. 50; den Boon et al., 2012 PNAS, ref. 51; Bari et al., 2011; Cellular and Molecular Life Sciences, ref. 52 These new references have been included in the revised manuscript). However, we have performed additional dose-dependence (0.01, 0.1, 1.0, 5.0 and 10 µM) studies with JWH133 by confirming no-cytotoxicity also when BV2 cells were treated with LPS in combination with JWH133 at the dose of 1 µM. The latter was also efficacious in increasing the IL-10 gene expression, as demonstrated by the new qRT-PCR experiment. These new results are now included in Supplementary materials.
The authors considered URB597 as a selective inhibitor of FAAH. However, for this latter mechanism, URB597 showed a maximal activity at 1 μM (Tanaka et al., 2019), the authors used the drug at much higher concentrations. The authors should discuss this discrepancy.
R: As reported in the literature, URB597 has been used in several studies as a selective FAAH inhibitor in the concentration range 100 nM - 10 μM, depending on the cellular models. As pointed out above, in BV-2 cells it was successfully used at 10 μM, and we also used the same concentration. See also reply to the above point 2.
The aim of this study looks unfocused. In particular, the use of URB597 induced an accumulation of anandamide, that is an endocannabinoid and also an endovanilloid. A better discussion of the possible involvement of CB1 and TRPV1 receptors on the crosstalk of the eCB and the S1P systems is needed.
R: In this work we mainly focused on CB2 because its expression was much higher than that of CB1 in inflamed microglia. Consistently, CB2 is known to represent the most relevant cannabinoid receptor in neuroinflammation (Scipioni et al., 2022 – ref. 3). As for TRPV1, its influence on microglia has been well-established in recent years (Scipioni et al., 2022 – ref. 3). For instance, it can regulate NLRP3 inflammasome activation and consequently mediate neuroinflammation (Zhang et al., 2021, ref.66 - this new reference has been included in the revised manuscript). Therefore, the increased eCB levels observed upon FAAH inhibition may be protective also by engaging TRPV1 and thus reducing inflammasome activation. This hypothesis has been included in the revised Discussion.
Minor issue:
Please change the number of figure on immune image (Fig. 2 vs Fig. 1).
R: Please note that in the Word file it is correct, the error was unexpectedly generated by PDF file conversion.
Please comment in Fig. 3 why SLP is not able to reduce significantly the increased mRNA levels of TNFα induced by LPS treatment.
R: The effect of SLP on TNFα is not significant although there is a trend towards decrease. Data (n=3 of three independent experiments) are statistically analysed using two-way Anova that compares groups and, regarding numerical data of TNFα expression, they are beyond the range of significance with respect to the other ones.
Please change the WB image of CB1.
R: Done as requested.

Reviewer 2 Report
In the present study, the authors investigated the anti-inflammatory effects of the interplay between sphingosine kinases and endocannabinoid signaling in BV2 mouse microglia cells and suggests a novel therapeutic tool for the treatment of neuroinflammatory diseases. They showed interesting data, with high relevance in advancing of knowledge in the field. The manuscript is well-written and easy to read.
Minor comments:
Figures
-it might help for the clarity of the figure, to put at the top of each panel the letters such as a), b), instead of at the bottom
-pg.5, it should be Figure 2, in the legend
-Figure 4, graphs legend missing
Author Response
Responses to Reviewer 2
In the present study, the authors investigated the anti-inflammatory effects of the interplay between sphingosine kinases and endocannabinoid signaling in BV2 mouse microglia cells and suggests a novel therapeutic tool for the treatment of neuroinflammatory diseases. They showed interesting data, with high relevance in advancing of knowledge in the field. The manuscript is well-written and easy to read.
Minor comments:
Figures
-it might help for the clarity of the figure, to put at the top of each panel the letters such as a), b), instead of at the bottom
-pg.5, it should be Figure 2, in the legend
R: We thank the Referee for her/his kind appreciation of our work, and for constructive suggestions on how to improve its clarity and impact. We have renumbered the figures as requested, and better marked them with letters at the top. Please note that the numbering of Figure 2 was correct in the Word file but was unexpectedly changed by PDF file conversion.
-Figure 4, graphs legend missing
R: Thank you for your suggestion that we have included in the revised manuscript.
Reviewer 3 Report
Based on experiments performed on BV2 microglia culture the authors suggest that inhibition of fatty acid hydrolase and activation of the CB2 receptor, respectively exert an anti-inflammatory effect by reducing the expression of cytokines TNFalpha, of IL-1 and of the sphingosine kinases 1/2. They also suggest that inhibition of sphingosine kinase 1/2 activity (by SLP7111228, SLM6031434, each at 1 µM) represses LPS-induced transcription of TNFalpha and IL-1. Thus, the authors believe sphingosine kinases mediate an inflammatory effect.
The data show that
(1) LPS increases the levels of transcripts for TNFalpha, for IL-1 and for the sphingosine kinases 1/2.
(2) URB597 (10 µM), an inhibitor of fatty acid hydrolase decreases transcripts for TNFalpha, IL-1 and SphK1/2 completely abolishing the increase due to LPS.
(3) Incubation with the cannabinoid JWH133 (1 µM) decreases the levels of transcripts for TNFalpha, IL-1 and SphK1/2 completely abolishing the increase due to LPS
(4) Incubation with sphingosine kinase inhibitors (SLP7111228, SLM6031434) decreases the levels of transcripts for TNFalpha and for IL-1 largely abolishing the increase due to LPS.
(5) LPS has no effect on the level of sphingosine-1-phosphate but causes a minute decrease in sphingosine levels.
(6) URB597 (10 µM) slightly decreases the level of sphingosine-1-phosphate (in non-LPS-treated cells).
(7) URB597 (10 µM) - but not incubation with LPS – increases levels of anandamide, of oleic acid ethanolamide, of cis-vaccenic acid ethanolamide, of palmitoylethanolamide, and of docosahexaenoyl ethanolamide, respectively.
A problem with interpreting data (2)-(4) is that they fail to offer unambiguous evidence for target-specific effects of the compounds (URB597, JWH133, SLP7111228, SLM6031434 or their combination) used for reverting the rise of transcript levels.
The changes in transcript levels are given as 2^(-ΔΔCT) values and thus are related to the commonly invariable levels of standard transcripts (18S, beta-actin, GAPDH). Nevertheless, 2^(-ΔΔCT) values do not allow for the interpretation that the effects of the compounds used for reverting the rise of transcript levels (URB597, JWH133, SLP7111228, SLM6031434 or their combination) are specific for their targets and well tolerated. Uncertainty Is due to the obvious caveat that incubation with LPS plus the rescue compounds (URB597, JWH133, SLP7111228, SLM6031434 or their combination) might cause harm to BV2 culture. Conversely, it is not a given that BV2 cells challenged for 4.5 hrs with LPS plus each of the compounds (URB597, with JWH133, with SLP7111228, SLM6031434 or their combination) are viable at the time of cell harvest.
Therefore, it is mandatory to rule out that the observed effects reflect selection of BV2 culture that is refractory to LPS-challenge. Possibly, resilient cells are selected during incubation whereas those susceptible to LPS succumb to treatment with LPS plus the rescue compounds (URB597, JWH133, SLP7111228, SLM6031434 or their combination). In fact, the data suggest some heterogeneity of the culture. The bar diagrams shown in Figs. 1, 3 and 6 reveal variable fold-increments in transcript levels for TNFalpha, IL-1 and SphK1 between experiments. This concern may be revoked if viability at the end of the incubation were at the level of controls.
If however viability had indeed been affected by LPS plus rescue treatment the authors would be requested to employ methods to experimentally separate competing effects (reversal of the LPS-effect vs. loss of cells). Several approaches appear feasible.
The reason for presenting Figure 5 is not obvious. It would be more helpful to confirm CB2 receptor-mediated effects with the use of a selective antagonist.
Additional questions
The incubation period with URB597, with JWH133, with SLP7111228, SLM6031434 or their combination should be stated explicitly: was the incubation continued after the addition of LPS?
Which TaqMan probes were used for the amplification of GAPDH and Beta-actin cDNA?
What was the origin of the Abcam #ab37977 antibody?
"Vehicles" (solvents) used in Figs. 1, 3 and 6. Were different vehicles used which may account for the variable change in mRNA levels of TNF, IL-1, SphK1? Please specify vehicle. If the assay conditions for ctr. and LPS were the same across experiments mean changes of 2^(-ΔΔCT) values from all three experiments should be given.
Figure 4: It appears inappropriate to depict error bars for the intra assay variability when the variation between experiments of lipid metabolite levels should be shown. Original data (MS signal traces used for quantification) should be included instead.
Author Response
Responses to Reviewer 3
Based on experiments performed on BV2 microglia culture the authors suggest that inhibition of fatty acid hydrolase and activation of the CB2 receptor, respectively exert an anti-inflammatory effect by reducing the expression of cytokines TNFalpha, of IL-1 and of the sphingosine kinases 1/2. They also suggest that inhibition of sphingosine kinase 1/2 activity (by SLP7111228, SLM6031434, each at 1 µM) represses LPS-induced transcription of TNFalpha and IL-1. Thus, the authors believe sphingosine kinases mediate an inflammatory effect.
The data show that
(1) LPS increases the levels of transcripts for TNFalpha, for IL-1 and for the sphingosine kinases 1/2.
(2) URB597 (10 µM), an inhibitor of fatty acid hydrolase decreases transcripts for TNFalpha, IL-1 and SphK1/2 completely abolishing the increase due to LPS.
(3) Incubation with the cannabinoid JWH133 (1 µM) decreases the levels of transcripts for TNFalpha, IL-1 and SphK1/2 completely abolishing the increase due to LPS
(4) Incubation with sphingosine kinase inhibitors (SLP7111228, SLM6031434) decreases the levels of transcripts for TNFalpha and for IL-1 largely abolishing the increase due to LPS.
(5) LPS has no effect on the level of sphingosine-1-phosphate but causes a minute decrease in sphingosine levels.
(6) URB597 (10 µM) slightly decreases the level of sphingosine-1-phosphate (in non-LPS-treated cells).
(7) URB597 (10 µM) - but not incubation with LPS – increases levels of anandamide, of oleic acid ethanolamide, of cis-vaccenic acid ethanolamide, of palmitoylethanolamide, and of docosahexaenoyl ethanolamide, respectively.
A problem with interpreting data (2)-(4) is that they fail to offer unambiguous evidence for target-specific effects of the compounds (URB597, JWH133, SLP7111228, SLM6031434 or their combination) used for reverting the rise of transcript levels.
The changes in transcript levels are given as 2^(-ΔΔCT) values and thus are related to the commonly invariable levels of standard transcripts (18S, beta-actin, GAPDH). Nevertheless, 2^(-ΔΔCT) values do not allow for the interpretation that the effects of the compounds used for reverting the rise of transcript levels (URB597, JWH133, SLP7111228, SLM6031434 or their combination) are specific for their targets and well tolerated. Uncertainty Is due to the obvious caveat that incubation with LPS plus the rescue compounds (URB597, JWH133, SLP7111228, SLM6031434 or their combination) might cause harm to BV2 culture. Conversely, it is not a given that BV2 cells challenged for 4.5 hrs with LPS plus each of the compounds (URB597, with JWH133, with SLP7111228, SLM6031434 or their combination) are viable at the time of cell harvest.
Therefore, it is mandatory to rule out that the observed effects reflect selection of BV2 culture that is refractory to LPS-challenge. Possibly, resilient cells are selected during incubation whereas those susceptible to LPS succumb to treatment with LPS plus the rescue compounds (URB597, JWH133, SLP7111228, SLM6031434 or their combination). In fact, the data suggest some heterogeneity of the culture. The bar diagrams shown in Figs. 1, 3 and 6 reveal variable fold-increments in transcript levels for TNFalpha, IL-1 and SphK1 between experiments. This concern may be revoked if viability at the end of the incubation were at the level of controls.
If however viability had indeed been affected by LPS plus rescue treatment the authors would be requested to employ methods to experimentally separate competing effects (reversal of the LPS-effect vs. loss of cells). Several approaches appear feasible.
R: Thank you for the detailed analysis.
The qRT-PCR graphs show the relative expression of each gene as 2^(-ΔΔCT): normalized to different housekeeping genes (18S, beta-actin, GAPDH) that are constitutively expressed in biological samples and normalized to control samples (untreated), in line with Livak and Schmittgeen, 2001, ref 81. The expression of housekeeping genes remained unchanged after different cell treatments as confirmed by the constant raw Ct value obtained by instrument analysis. This seems to be a largely used, well consolidated procedure that allows accurate quantification of mRNA, as demonstrated by several studies, ours included (Di Meo et al., 2022 IJMS; Rapino et al., 2019; Molecules.;24(7):1432).
In order to rule out that the observed effects of URB597 and JWH133 reflect selection of BV2 culture refractory to LPS-challenge, we have performed additional dose-dependence (0.01, 0.1, 1.0, 5.0 and 10 µM) studies with URB597 and JWH133 by confirming no-cytotoxicity induced by both compounds also when BV2 cells were treated with LPS in combination with URB597 or JWH133 at the doses of 10 µM and 1.0 µM, respectively. These new results are now included in Supplementary materials (Figure S1 and S3).
The reason for presenting Figure 5 is not obvious. It would be more helpful to confirm CB2 receptor-mediated effects with the use of a selective antagonist.
R: Figure 5 is shown in order to confirm the strong expression of the CB2 receptor on protein level in BV2 cells. The figure furthermore shows that treatment with URB597 (at least under the conditions applied here) did not downregulate the CB2 receptor protein but, in contrast, significantly increased it (as mentioned in the Results section). We think that this is relevant information.
We agree that the use of antagonists could possibly be helpful. Nevertheless, our data with the agonist show that CB2 receptor activation fully mimicked the effects of URB597 as studied here. It is important to consider that CB2 is a lipid GPCR that is activated upon URB597 treatment via an endogenous autocrine lipid. In this setting, lipophilic agonists may not always be fully displaced by exogenous antagonists. For example, the crystal structure of the lipid GPCR, S1P1, revealed that the endogenous agonist accesses the receptor from within the plasma membrane via lateral diffusion (see Parrill et al. 2012, Sci Signal Vol 5 Issue 225 pe23). Therefore, the use of antagonists for studying autocrine signalling of endogenous lipids is not straightforward and may cause high data variability. This is why we preferred the use of an agonist in the present study. The agonist has the additional advantage that it indicates a potential therapeutic strategy alternatively to URB597.
Additional questions
The incubation period with URB597, with JWH133, with SLP7111228, SLM6031434 or their combination should be stated explicitly: was the incubation continued after the addition of LPS?
R: We have improved the Materials and Methods section by adding the requested details (page 11-12, lines 373-377).
Which TaqMan probes were used for the amplification of GAPDH and Beta-actin cDNA?
R: To amplify GAPDH and β-actin were used primers and the SensiFASTTM SYBR Lo-ROX kit, as clarified in the Materials and Methods section (page 12, lines 399-406) of revised manuscript.
What was the origin of the Abcam #ab37977 antibody?
R: It was a rabbit polyclonal from Abcam (Cambridge, UK).
"Vehicles" (solvents) used in Figs. 1, 3 and 6. Were different vehicles used which may account for the variable change in mRNA levels of TNF, IL-1, SphK1? Please specify vehicle. If the assay conditions for ctr. and LPS were the same across experiments mean changes of 2^(-ΔΔCT) values from all three experiments should be given.
R: DMSO was the vehicle used for all experiments, at the same dilutions used for the various compounds. This point has been clarified in the Materials and Methods section (page 12, lines 375-377) of revised manuscript.
Figure 4: It appears inappropriate to depict error bars for the intra assay variability when the variation between experiments of lipid metabolite levels should be shown. Original data (MS signal traces used for quantification) should be included instead.
R: We performed two experiments, both of them in triplicates, that showed nicely overlapping results. Obviously, we could not perform statistics on results from two experiments and found it more appropriate to test the intra-assay variability separately for the two experiments.
Round 2
Reviewer 1 Report
This manuscript has been further improved by the revision.
Author Response
Thank you for appreciating the revised manuscript.
Reviewer 3 Report
Comment on the revised manuscript
Figure 4: The authors' reply that they did not run statistical tests on the assay results in Figure 4 is misleading. It is as unnecessary as it is inappropriate to show p-values for differences between results obtained from two samples. This ought to be corrected.
Figure 5: It is beyond doubt that the use of a CB2-receptor antagonist to prevent the effect of JWH133 would be rather helpful in substantiating the claim that CB2-receptors exert an anti-inflammatory effect in BV2-cells.
Author Response
"Figure 4: "It is as unnecessary as it is inappropriate to show p-values for differences between results obtained from two samples. This ought to be corrected".
Answer: With due respect we want to point out that we made statistics of triplicate values, not of two samples, as explained in the Legend to Figure 4.
"Figure 5: "It is beyond doubt that the use of a CB2-receptor antagonist to prevent the effect of JWH133 would be rather helpful in substantiating the claim that CB2-receptors exert an anti-inflammatory effect in BV2-cells." Answer: We think that the results with the specific agonist, JWH133, clearly show that the CB2 receptor exerts an anti-inflammatory effect in BV2 cells, and an antagonist will not add additional information here. The antagonist, however, would help to strengthen the hypothesis that CB2 is involved in an autocrine loop triggered by the FAAH inhibitor. As pointed out in our previous reply, it is difficult to block autocrine loops of endogenous lipids with exogenous antagonists. Therefore, we have discussed this cautiously in this paper. Taken together, we think that the anti-inflammatory activity of the CB2 receptor, shown also previously by other research groups, is sufficiently supported by our data with JWH133. We hope that we can convince the reviewer that experiments with CB2 antagonists will not add additional information to the data with JWH133, and will not be straightforward in combination with the FAAH inhibitor.